# The Value of Delta Middle Cerebral Artery Peak Systolic Velocity for the Prediction of Twin Anemia-Polycythemia Sequence—Analysis of a Heterogenous Cohort of Monochorionic Twins

**DOI:** 10.3390/jcm11247541

**Published:** 2022-12-19

**Authors:** Anthea de Sainte Fare, Ivonne Bedei, Aline Wolter, Johanna Schenk, Ellydda Widriani, Corinna Keil, Siegmund Koehler, Franz Bahlmann, Brigitte Strizek, Ulrich Gembruch, Christoph Berg, Roland Axt-Fliedner

**Affiliations:** 1Department of Prenatal Medicine and Fetal Therapy, University Hospital Giessen, 35392 Giessen, Germany; 2Department of Prenatal Medicine and Fetal Therapy, University Hospital Marburg, 35041 Marburg, Germany; 3Department of Ultrasound Diagnostics and Prenatal Medicine, Buergerhospital Frankfurt, 60318 Frankfurt, Germany; 4Department of Obstetrics and Prenatal Medicine, University Hospital Bonn, 53127 Bonn, Germany; 5Department of Prenatal Medicine and Gynaecological Ultrasound, University Hospital Cologne, 50937 Cologne, Germany

**Keywords:** monochorionic-twins, twin-anemia-polycythemia sequence, twin-to-twin transfusion syndrome, delta MCA-PSV

## Abstract

Introduction: Twin anemia-polycythemia sequence (TAPS) is a complication in monochorionic-diamniotic (MCDA) twin pregnancies. This study analyzes whether the prenatal diagnosis using delta middle cerebral artery-peak systolic velocity (MCA-PSV) > 0.5 multiples of the median (MoM) (delta group) detects more TAPS cases than the guideline-based diagnosis using the MCA-PSV cut off levels of >1.5 and <1.0 MoM (cut-off group), in a heterogenous group of MCDA twins. Methods: A retrospective analysis of 348 live-born MCDA twin pregnancies from 2010 to 2021 with available information on MCA-PSV within one week before delivery and hemoglobin-values within 24 h postnatally were considered eligible. Results: Among postnatal confirmed twin pairs with TAPS, the cut-off group showed lower sensitivity than the delta group (33% vs. 82%). Specificity proved higher in the cut-off group with 97% than in the delta group at 86%. The risk that a TAPS is mistakenly not recognized prenatally is higher in the cut-off group than in the delta group (52% vs. 18%). Conclusions: Our data shows that delta MCA-PSV > 0.5 MoM detects more cases of TAPS, which would not have been diagnosed prenatally according to the current guidelines. In the collective examined in the present study, TAPS diagnostics using delta MCA-PSV proved to be a more robust method.

## 1. Introduction

Monochorionic diamniotic (MCDA) pregnancies represent the most common subgroup of monogynous twins, with approximately 65% [1]. A specific risk in MCDA pregnancies is the presence of vascular anastomoses across the common placenta. In 10% of MCDA fetuses, pregnancy is complicated by a twin-to-twin transfusion syndrome (TTTS), in 5% by a twin anemia-polycythemia sequence (TAPS) [2,3].

TAPS is a subtype of TTTS in MCDA twin pregnancies. The pathophysiological precondition of TAPS are tiny arterio-venous (AV) anastomoses (<1 mm) in the placenta, which connect the blood circulation between the twins [4]. This clinical picture can arise spontaneously or as a result of TTTS laser therapy (13%) after incomplete laser coagulation, as well as through recanalization of the coagulated anastomoses [5]. In contrast to classical TTTS, in which hypervolemia and hypovolemia dominate pathophysiologically due to a strong unbalanced net blood flow, in TAPS tiny anastomoses enable only a minimal but continuous unbalanced net intertwin flow of blood. This results in a large intertwin hemoglobin difference, but not in relevant hyper and hypovolemia [5]. Similar to the TTTS, there is a donor and a recipient twin. The donor twin shows anemia due to the loss of red blood cells to the recipient, whereas the recipient twin shows polycythemia due to the excessive transmission in red blood cells [6]. For the antenatal diagnosis of TAPS, the peak systolic velocity of the middle cerebral artery (MCA-PSV) is determined by Doppler measurements [7]. The criteria previously used to indicate a antenatal TAPS diagnosis according to the guidelines were the recipients MCA-PSV < 1.0 multiples of the median (MoM) and the donors MCA-PSV > 1.5 MoM, as well as the absence of amniotic fluid abnormalities [8]. Postnatal diagnostic criteria for TAPS are an intertwin hemoglobin difference >80 g/L, evidence of intertwin anastomoses in the placenta or a reticulocyte index of over 1.7 [8,9].

Various researchers recently proposed new cut-off values for MCA in TAPS diagnostics. This shows the urgency of revising the currently valid MCA cut-off values in TAPS diagnostics. After a retrospective data analysis with 154 monochorial-diamniotic twin pairs, Tavares de Sousa et al. even advocated setting the cut-off value of the delta MCA-PSV at 0.373 MoM [10].

Khalil et al. proposed that the best TAPS diagnosis can be made if the cut-off values are set at MCA-PSV > 1.5 MoM for the donor twin and <0.8 MoM for the recipient or delta MCV-PSV > 0.8 MoM [11].

Fishel-Bartal et al. [12] already observed the positive correlation between a high difference in the flow velocity of the middle cerebral artery (delta MCA-PSV) in the twins and a high hemoglobin difference measured postnatally. This hypothesis was tested by the University of Leiden in 2019 and confirmed with the same result [13]. It was also considered that this investigation method represents a better indicator for the antenatal detection of TAPS. A delta MCA-PSV value of more than 0.5 MoM was concluded to be pathological [13].

However, both studies were only carried out on a small group of patients. The aim of our study was to analyze the diagnostic potential of delta MCA-PSV > 0.5 MoM in a large group of heterogenous MCDA twin pairs.

## 2. Methods

This study is a retrospective data analysis in five tertiary referral centers in Germany attending a high number of twin pregnancies. All MCDA twin pregnancies from January 2010 to January 2021 were considered. Both antenatal and postnatal values were taken into account. The study included 348 MCDA twin pregnancies from the following German prenatal diagnostic centers: Justus Liebig University Giessen, Philipps University Marburg, Buergerhospital Frankfurt, Rheinische Friedrich-Wilhelm’s University Bonn, and University of Cologne. 

All perinatal medicine specialists participating were informed whether therapeutic interventions such as laser treatment were performed on the fetuses.

Every twin pair underwent detailed anomaly survey including sequential fetal echocardiography at the 18th–22nd weeks of gestation by experienced level 3 operators according to the DEGUM (Deutsche Gesellschaft fuer Ultraschall in der Medizin) guidelines [7]. Cardiac findings in TAPS twins included atrioventricular valve regurgitation, and/or functional pulmonary stenosis or atresia and/or a cardiothoracic ratio ≥0.5.

The confidence interval (CI) was determined to check the precision of the result.

Cases in which antenatal Doppler measurements of the MCA of both twins were available, as well as postnatal information about the hemoglobin values, were considered eligible. Cases with MCA Doppler longer than one week prior to delivery and those in which the postnatal hemoglobin value was not determined within 24 h of the time of birth were excluded. The postnatal diagnosis of TAPS was based on an intertwin hemoglobin difference >80 g/L. Postnatally, the twin pairs were divided into two groups:Control group—without TAPS (postnatal hemoglobin difference of <79 g/L);TAPS (hemoglobin-difference of >80 g/L).

The antenatal TAPS diagnosis was determined by ultrasound Doppler measuring MCA-PSV and divided into three groups:Control group—with normal MCA-PSV values;Twins who exceeded the cut-off values MCA-PSV > 1.5 and <1.0 MoM;Twins with delta MCA-PSV > 0.5 MoM.

Statistical analysis was performed using SSPS-21 (IBM, Armonk, NY, USA). Multiples of median (MoM) were determined in dependence on the gestational age using the “Perinatology calculator” [14]. The nominal and ordinal scaled data were examined using the chi-square test and the differences between the two groups were shown in a cross table. In case the chi-square test could not be used to check the significance due to a rule violation, the Fisher test (exact chi-square test) was used. The statistical quality criteria sensitivity, specificity, positive and negative predictive values were calculated using a 2 × 2 table in order to determine the value of the respective tests. The confidence intervals were set to refine the results.

To test the strength of the correlation, a Pearson test was performed on the normally distributed data. Spearman’s correlation coefficient was used to check the correlation of the metrically distributed data.

As a secondary question, it was checked if there was a more accurate prenatal delta MCA-PSV cut-off than the one used of 0.5 MoM by using a receiver operating statistics curve (ROC). The significance level of the *p*-value was set at 0.05 (5%). The Bonferroni correction was used to neutralize alpha accumulation in multiple tests [15].

The work was approved by the Ethics Committee Giessen (ID: 254/19).

## 3. Results

This study includes 348 live-born MCDA twin pairs. Of the 348 twin pairs, 95 were recruited at the University Hospital Giessen, 91 at the University Hospital Marburg, 73 at the Buergerhospital Frankfurt, 53 at the University Hospital Cologne, and 36 at the University Hospital Bonn.

A total of 288 twin pairs in the control group (83%) had a hemoglobin difference of less than 79 g/L postnatally; 60 twin pairs indicated a TAPS with a postnatal hemoglobin difference >80 g/L.

TAPS is a known complication after laser coagulation for TTTS. Data analysis shows that 45% (*n* = 27) of TAPS cases emerged after laser coagulation for TTTS. In 55% (*n* = 33) of the TAPS cases, the clinical picture emerged spontaneously. The mean maternal age of the study participants was 32 years (range 17 to 45 years).

A total of 258 twin pairs showed no abnormalities in MCA-PSV (control group). 29 twin pairs had an MCA-PSV > 1.5 MoM in one twin and <1.0 MoM in the other twin, and thus fulfilled the conditions of an antenatally diagnosed TAPS according to the current criteria. Of the cases, 90 showed a delta MCA-PSV > 0.5 MoM. The 90 twin pairs were composed of the 29 twin pairs detected by MCA-PSV cut-off level of >1.5 and <1.0 MoM method and 61 additional twin pairs detected as TAPS by the delta method only.

Both Figure 1 and Table 1 show the relationship between the hemoglobin difference and the MCA-PSV. The diagnostic criteria delta MCA-PSV > 0.5 MoM vs. MCA-PSV cut-off level of >1.5 and <1.0 MoM were analyzed regarding to the postnatal hemoglobin difference. Of 29 twin pairs diagnosed antenatally as TAPS based on the current criteria (MCA-PSV cut-off level of >1.5 and <1.0 MoM), 20 of the twin pairs (69%) were confirmed postnatally by an intertwin hemoglobin difference of >80 g/L. Of the 90 twin pairs diagnosed antenatally as TAPS due to delta MCA-PSV > 0.5 MoM, in 49 (54%) cases the diagnosis of TAPS was postnatally confirmed. Among the 61 additional TAPS cases suspected by the delta method, 29 (48%) showed postnatally a TAPS. A total of 11 twin pairs with TAPS were not detected prenatally neither by one nor the other diagnostic method. Of 60 TAPS cases, which were confirmed postnatally, the delta MCA-PSV > 0.5 MoM method would have detected 29 twin pairs that exceeded the cut off values of MCA-PSV cut-off level of >1.5 and <1.0 MoM, and 20 additional cases that would not have been diagnosed by using MCA-PSV cut-off level of >1.5 and <1.0 MoM. In total, 49 of the 60 cases would have been discovered by delta MCA-PSV > 0.5 MoM prenatally. Diagnosis by using MCA-PSV cut-off level of >1.5 and <1.0 MoM would have detected 29 of the 60 TAPS cases prenatally. A proportion of 46% (41 of 90) of cases discovered by delta MCA-PSV > 0.5 MoM were incorrectly diagnosed prenatally as TAPS (vs. 31%, 9 of 29) cases by MCA-PSV cut-off level of >1.5 and <1.0 MoM). A positive correlation between the hemoglobin difference and the MCA-PSV could be shown, both measured in delta MCA-PSV > 0.5 MoM and in the absolute MCA-PSV cut-off level of >1.5 and <1.0 MoM. This relationship turned out to be significant (*p* << 0.001).

The diagnostic quality of both antenatal TAPS diagnostic procedures is illustrated in Table 2.

Of the twin pairs actually suffering from TAPS, 49 of 60 (82%) were also diagnosed as well by using delta MCA-PSV > 0.5 MoM (sensitivity). The CI was set at 95%, which in the case of the sensitivity means that with a certainty of 95% the real values of the sensitivity are within the interval of 78% and 86%. In contrast, 86% (95% CI: 82–90%) of the MCDA pregnancies were correctly detected as not affected (specificity). In 54% (95% CI: 49–59%) of the cases, the twin pairs diagnosed antenatally as TAPS were also confirmed postnatal as TAPS (positive predictive value, PPV). The negative predictive value for diagnostics using delta MCA-PSV > 0.5 MoM is 96% (95% CI: 94–98%). In total, 49 of the 60 TAPS cases, which were confirmed postpartum would have also been detected antenatally with this diagnostic method. However, 11 cases still remained undetected antenatally, even with this method.

The diagnostic quality of the MCA-PSV cut-off level of >1.5 and <1.0 MoM, which is currently used in antenatal TAPS diagnostics according to the guidelines, is also shown in Table 2 [8,10]. The sensitivity of these current valid diagnostic criteria is 33% (95% CI: 28–38%) and the specificity is 97% (95% CI: 95–99%). In 69% (95% CI: 64–74%) of the cases, the TAPS twins diagnosed by means of MCA-PSV cut-off level of > 1.5 and < 1.0 MoM actually had TAPS (PPV). The negative predictive value for this method of antenatal diagnosis is 87% (95% CI: 84–91%).

Table 3 analyzes whether various fetal and neonatal characteristics are present or reduced for the respective diagnostic criterion. In this presentation, the delta MCA-PSV > 0.5 MoM considered only those 61 cases which would not have been detected by MCA-PSV cut-off level of >1.5 and <1.0 MoM. A total of 11 TAPS cases remained undetected prenatally with both methods. The data analysis shows that the twins who were diagnosed antenatally as TAPS due to MCA-PSV cut-off level of >1.5 and <1.0 MoM, were born earlier on average than those who were diagnosed using delta MCA-PSV > 0.5 MoM (29th vs. 32nd week of gestational age at birth). Furthermore, the data indicate a connection between the causality of TAPS (spontaneous vs. post-laser TAPS) and the TAPS diagnostic procedure. The post-laser TAPS cases, which were diagnosed as TAPS due to the absolute cut-off values (MCA-PSV cut-off level of >1.5 and <1.0 MoM), represent a larger proportion with 48% (15 twin pairs) than the TAPS cases, which were diagnosed using delta MCA-PSV > 0.5 MoM (15 twin pairs, 25%).

The eleven undetected TAPS cases had an average hemoglobin difference of 110 g/L postnatally. The gestational age of 34 weeks was higher than in the prenatally diagnosed TAPS cases. The average delta MCA-PSV in this group was 0.3 MoM. The difference in birth weight of the twins in this group was 433 g (22%) on average, higher than that of the prenatally diagnosed TAPS cases.

Cardiac findings typical of TAPS occurred more frequently in the TAPS cases, which were diagnosed due to the absolute cut-off values, than in the cases which were diagnosed due to delta MCA-PSV > 0.5 MoM (72% vs. 42%). There was no association between postnatal diagnosis of TAPS and either amniotic fluid abnormalities or weight discrepancies of the twin pairs. The significantly higher intertwin hemoglobin difference of 108 g/L (vs. 75 g/L) in the twins diagnosed antenatally with MCA-PSV cut-off level of >1.5 and <1.0 MoM is noticeable (*p* << 0.001). All these data refer to cases that were prenatally indicative of TAPS and not to postnatally confirmed TAPS cases.

In this study the precision of the delta MCA-PSV value of 0.5 MoM is evaluated by using a ROC curve as a secondary question, which is shown in Figure 2.

To detect an optimal delta MCA-PSV cut-off value, the Youden index (sensitivity + specificity-1) was used. The optimal cut-off value is defined at a maximum high sensitivity and a lowest possible false positive value (1-specificity) [16]. With a sensitivity of 0.817 (81.7%) and a 1-specificity of 0.142 (14.2%), the most accurate delta MCA-PSV cut-off value in this collective was detected at 0.45 MOM. The area under the curve (AUC) is 0.889 (95% CI: 0.936–0.843).

## 4. Discussion/Conclusions

This study was conducted to evaluate the diagnostic accuracy of delta MCA-PSV in a heterogenous group of MCDA twin gestations regarding the presence of twin anemia polycythemia sequence (TAPS) postnatally. In 49 confirmed cases of TAPS twin pairs in this study delta MCV-PSV > 0.5 MoM turned out to be a higher diagnostic accuracy in terms of sensitivity and comparable specificity compared to the currently implemented MCA-PSV cut-off level of >1.5 and <1.0 MoM for diagnosis of antenatal TAPS in MCDA twin gestations.

The parameter delta MCA-PSV > 0.5 MoM achieved satisfactorily high sensitivity and specificity for the prediction of TAPS. Particularly noteworthy is the high negative predictive value of 96% for this criterion. In contrast, the standard definition of TAPS with MCA-PSV cut-off level of >1.5 and <1.0 MoM was shown to have an inferior sensitivity. Based on the delta MCA-PSV > 0.5 MoM cut-off, only 18% (11/60) of confirmed postnatal TAPS cases were missed, in contrast to 52% (31/60) with MCA-PSV cut-off level of >1.5 and <1.0 MoM. 

In summary, Table 2 shows that the two methods are completely adjacent and significantly different in terms of sensitivity, specificity, positive and negative predictive value. The delta MCA-PSV method is far better at spotting those that are actually healthy, and the MCA-PSV absolute value method is far more effective at finding cases affected by TAPS. This means that the value of these methods mainly depends on the research question. If the primary goal is to filter out the affected cases precisely and with the lowest possible false positive rate, the absolute MCA-PSV method should be chosen. However, one should be aware that some cases remain undetected, so that appropriate diagnostics and therapy are not carried out and the children may grow up with permanent damage. However, if as many cases as possible are to be detected prenatally, even with the risk of a higher false positive rate, the delta MCA-PSV method must be chosen. It is certainly more important to detect cases that are very likely to be affected as early as possible, in order to observe further diagnosis in small steps and to be able to intervene as quickly as possible in event of aggravation.

Thus, the use of delta MCA-PSV > 0.5 MoM could clarify prenatal TAPS diagnosis and help affected twins benefit from more intensive prenatal surveillance and intervention, if appropriate.

The present results are consistent with the recently reported results of Tollenaar et al. [13], who presented results on the improved prediction of TAPS by delta MCA-PSV in a selected group of MCDA twin pregnancies with a confirmed diagnosis of TAPS postnatally. They reported in a series of 35 twin pairs with postnatally confirmed TAPS a sensitivity of 46% and 100% specificity for MCA-PSV cut-off level of >1.5 and <1.0 MoM compared to an 83% sensitivity and 100% specificity for delta MCA-PSV > 0.5 MoM. Of note, they reported on 13 twin pairs of the 35 confirmed TAPS cases postnatally with a delta MCA-PSV > 0.5 MoM but not reaching the MCA-PSV cut-off level of >1.5 and <1.0 MoM criterion; nine donors and four recipients of these 13 TAPS cases presented with normal MCA-PSV values. These results were considered in our study. Of the 60 TAPS cases, confirmed postnatally, 20 cases were prenatally discovered only by using delta MCA-PSV > 0.5 MoM. Diagnosis by MCA-PSV cut-off level of >1.5 and <1.0 MoM would have left these cases undetected prenatally. Previous reports found higher hemoglobin level sensitivity values for MCA-PSV cut-off values of >1.5 and <1.0 MoM in a general population of MCDA pregnant woman. However, this does not apply in a specific subgroup of TAPS twin pairs [12]. Specifically, this study and the recent study of Tollenaar et al. did not examine postnatal hemoglobin levels as the primary outcome, but rather the hemoglobin difference between the twins, which is the basis of a postnatal TAPS diagnosis [13].

With regard to perinatal outcomes, TAPS twins diagnosed by MCA-PSV cut-off level of >1.5 and <1.0 MoM were born at a significantly lower gestational age compared to those diagnosed by delta MCA-PSV > 0.5 MoM (29 weeks vs. 32 weeks, respectively, *p* < 0.001). In the group of MCA-PSV cut-off level of >1.5 and <1.0 MoM there were more intrauterine interventions compared to the delta MCA-PSV > 0.5 MoM twin pairs, with a statistical difference. It might be argued that since this was a retrospective analysis, the latter group was considered to be mildly or moderately affected TAPS pairs and therefore the decision to deliver was a conscious one. Consistent with this argument, in our study, twin pairs diagnosed by using delta MCA-PSV showed a lower Hb difference (75 g/L vs. 108 g/L) and thus lower expression of TAPS than the twin pairs diagnosed by cut-off MCA-PSV. This statement seems to contradict the data of Tollenaar et al., who found no significant differences in gestational ages at delivery between both groups, however, they also reported later gestational ages in the group of delta MCA-PSV > 0.5 MoM compared to MCA-PSV cut-off level of >1.5 and <1.0 MoM (34 weeks vs. 31 weeks of gestation). Moreover, in our study we reported a high proportion of cardiac findings in both groups (42% and 72%, respectively), probably reflecting a more advanced stage of disease (Stage ≥ 2 according to the current guidelines [8]), and therefore a lower gestational ages might be attributed to the fact of more severe stage of disease. Tollenaar et al. did not report on cardiac disease in their study.

An important finding of our analysis is the significantly lower hemoglobin and hematocrit difference in the twins detected prenatally as TAPS by the delta MCA-PSV > 0.5 MoM compared with those detected by MCA-PSV >1.5 and <1.0 MoM. This may indicate that the cut-off group primarily captures the cases with severe TAPS. This translates into significantly higher delta MCA-PSV values in the group of MCA-PSV cut-off level of >1.5 and <1.0 MoM.

Tollenaar et al. have established a modified antenatal TAPS staging. According to this classification, no stage 1 TAPS was detected by the guideline-based prenatal diagnosis with MCA-PSV cut-off level of >1.5 and <1.0 MoM. With TAPS diagnostics using delta MCA-PSV > 0.5 MoM, 50% were already diagnosed at stage 1. Intervention such as intrauterine transfusion should be considered from stage 2. However, it should be mentioned at this point that up to now there is no standard and verified therapy after the 28th week of gestation for TAPS twins. The indication for delivery is a pregnancy beyond the 32nd week, plus the presence of a stage ≥ 3. In conclusion, prenatal TAPS diagnosis using delta MCA-PSV > 0.5 MoM seems to detect earlier TAPS stages than MCA-PSV cut-off level of >1.5 and <1.0 MoM [17].

Neither a significant discrepancy in birth weight nor differences in amniotic fluid was found between both groups, arguing that selective intrauterine growth restriction had no impact on the intertwin hemoglobin difference observed in our cohort. The Leiden group, however, reported on a higher birth-weight discordance in the TAPS group with delta MCA-PSV > 0.5 MoM, however both birthweight discordance in % and the rate of birth-weight discordance > 20% was not reported to be statistically different in this report [13].

In this study, it was found that the ideal delta MCA-PSV value is 0.45 MoM (Figure 2). Beyond this value, the number of incorrectly diagnoses increases more rapidly than the sensitivity, so that a further reduction of the cut-off value would not bring any positive benefit. Thus, according to our data, delta MCA-PSV of 0.45 MoM represents the value at which the diagnosis of TAPS should be considered. It was decided to set 0.5 MoM as cut-off value despite the small deviation from the calculated optimal MCA-PSV value (0.45 MoM), as this keeps this study comparable to the study by Tollenaar et al. on this topic [13]. Furthermore, the analysis with the modified delta value of 0.45 MoM showed no difference in data interpretation; both with a cut-off at 0.45 and at 0.5 MoM, 90 TAPS cases were diagnosed prenatally. Hence, with this dataset we were able to further support the previously used delta value of 0.5 MoM [13] and add another study that demonstrates the accuracy of this value. Additional studies attempted to identify an optimal delta cut-off value. In a study, Liu et al. compared the available prenatal diagnostic methods in terms of incidence, progression, and intervention rates in the presence of TAPS [18]. Considered were the currently valid criteria (>1.5/<1.0 MoM), the value of 0.373 MoM recommended by Tavares De Sousa, the values established by a Delphi procedure (>1.5/>0.8 MoM or delta MCA-PSV > 1.0 MoM), and the delta MCA-PSV value > 0.5 MoM advocated by Tollenaar et al. in 2019 It was found that the different diagnostic criteria revealed significant differences in the incidence of TAPS, severity, and prenatal intervention. The TAPS cases identified by the Delphi method, as well as the TAPS cases detected by the currently valid criteria, diagnosed prenatally mainly the cases with the highest rates of progression and intervention, showing the lowest incidence. Delta MCA-PSV > 0.373 MoM had the highest incidence and revealed mainly low-grade TAPS cases with a low rate of progression.

The question is which value represents the optimal diagnostic value. This depends, as mentioned above, on what is important for the investigator. If the aim is to filter out as many cases as possible prenatally at the risk of overdiagnosis, the delta values of <0.5 MoM should be applied. If the focus is on the most precise prenatal diagnosis with a high specificity, the currently valid criteria or those from the Delphi method should be used. 

According to Liu et al., this comparison of the currently available prenatal TAPS diagnostic options does not allow a definitive statement or standardization of the prenatal TAPS diagnostic criteria [18]. Further large studies are needed to determine the best possible cut-off value to achieve the highest possible diagnostic accuracy with the best possible neonatal outcomes.

A limitation of this retrospective study is the lack of histopathological studies on placental anastomosis, due to the inclusion of five different centers. An advantage of this study is that we analyzed a heterogenous group of MCDA twin pregnancies and not just a group of MCDA twins with postnatally confirmed diagnosis of TAPS. It is clear from our data that a prospective evaluation of MCDA twin pregnancies is warranted and intervention criteria and outcome parameters must be addressed. Another limitation is the definition of TAPS based on the postnatal hemoglobin difference. Different techniques of cord section, sometimes even consciously with a known diagnosis of TAPS, fluid applications, and different timings of postnatal blood sampling can lead to changes in hemoglobin concentrations and thus the intertwin hemoglobin difference.

In conclusion, from our data in a heterogenous group of MCDA twin pregnancies delta MCA-PSV > 0.5 MoM has a greater diagnostic accuracy for predicting TAPS compared to the MCA-PSV cut-off level of >1.5 and <1.0 MoM. Likewise, the earlier diagnosis of TAPS by delta MCA-PSV could lead to a more intense surveillance including fetal interventions, if indicated, of these pregnancies. Physicians involved in care of these high-risk twin pregnancies should be aware of these findings.

Finally, the question arises to what extent prenatal TAPS diagnostics can benefit from the results obtained. By diagnosing TAPS early and adjusting the prenatal observation accordingly, the progression of the disease can be detected as early as possible and, if necessary, mediated. Delta MCA-PSV does not imply any additional diagnostic effort and no diagnostic investments for the examiner. Only the evaluation differs from the currently used diagnostic method based on the absolute flow velocity of the ACM. An earlier diagnosis does not necessarily mean an earlier intervention but can sensitize the examiner and the patient to recognize a deterioration as early as possible.

The extent to which the outcome of the twins could improve if delta MCA-PSV > 0.5 MoM was diagnosed earlier needs to be controlled in further studies, preferably prospective studies. This cannot be adequately assessed in this study, since the prenatal diagnosis of TAPS disease with delta MCA-PSV > 0.5 MoM was not recognized by the prenatal diagnostician in most cases.

## Figures and Tables

**Figure 1 jcm-11-07541-f001:**
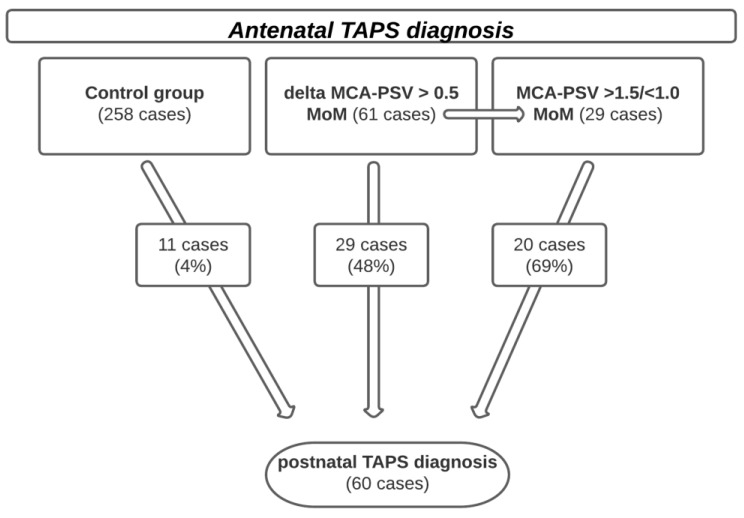
Shows the prenatal distribution of cases among the three diagnostic groups: the control group, the cases diagnosed as TAPS according to the criteria delta MCA-PSV > 0.5 MoM, and the cases that correspond to the diagnosis of TAPS according to the current guidelines (MCA-PSV >1.5/<1.0 MoM).

**Figure 2 jcm-11-07541-f002:**
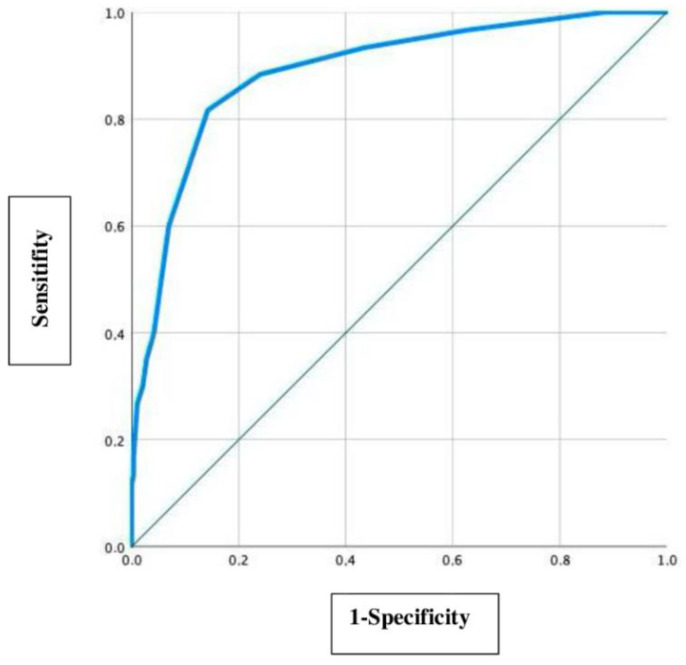
ROC curve to find the optimal MCA-PSV cut-off value.

**Table 1 jcm-11-07541-t001:** Correlation between prenatal TAPS diagnosis and postnatal TAPS confirmation.

		Postnatal Diagnosis of TAPS (Hb diff. >80 g/L)			
		Yes	No	Total	*p*-Value
**Prenatal**	Delta MCA-PSV >0.5 MoM	49 (48%)	41 (52%)	90 (100%)	<<0.001
	MCA-PSV > 1.5/<1.0 MoM	20 (69%)	9 (31%)	29 (100%)	<<0.001
	undetected TAPS cases	11 (4%)	247 (96%)	258 (100%)	<<0.001

A *p*-value < 0.05 was assumed to be statistically significant. MoM, multiples of the median; Hb diff, Hemoglobin difference; MCA-PSV, middle cerebral artery peak systolic velocity.

**Table 2 jcm-11-07541-t002:** Quality of antenatal diagnostic procedures.

	Antenatal TAPS Diagnostic	
	Delta MCA-PSV > 0.5 MoM (*n* = 90)	MCA-PSV >1.5/<1.0 MoM (*n* = 29)
Sensitivity Confidence interval (CI)	(49/60), 82% 78–86%	(20/60), 33% 28–37.9%
Specificity Confidence interval (CI)	(247/288), 86% 82.3–89.6%	(279/288), 97% 95.2–98.8%
Positive predictive value Confidence interval (CI)	(49/90), 54% 48.7–59.2%	(20/29), 69% 64.1–73.9%
Negative predictive value Confidence interval (CI)	(234/245), 96% 94–98%	(279/319), 87% 83.5–90.5%
False positive rate	(41/90), 46%	(9/29), 31%
False negative rate	(11/60), 18%	(31/60), 52%

MoM, multiples of the median; MCA-PSV, middle cerebral artery peak systolic velocity.

**Table 3 jcm-11-07541-t003:** Presentation of various fetal and neonatal characteristics in TAPS twins in relation to the different diagnostic criteria (delta MCA-PSV > 0.5 MoM, MCA-PSV >1.5/<1.0 MoM, and undiscovered TAPS cases prenatally).

	Antenatal TAPS Diagnostic				
Characteristic	delta MCA-PSV >0.5 MoM (n = 61)	MCA-PSV >1.5/<1.0 MoM (n = 29)	Undiscoverd TAPS (n = 11)	*p*-Value	
Sex of twin pairs (f/m)	31/30	14/15	4/7	0.896	
Gestational age at birth (weeks)	32	29	34	<0.001	
Birth-weight discordance (g)	361	332	433	0.6	
Type of TAPS:					
- Spontaneous	46 (75%)	14 (52%)	8 (78%)		
- Post-laser	15 (25%)	15 (48%)	3 (27%)	0.003	
Intertwin Hb discordance (g/L)	75	108	110	<<0.001	
Hk discordance	0.2	0.32	0.34	<<0.001	
Delta MCA-PSV (MoM)	0.57	1.01	0.3	<<0.001	
Amniotic fluid anomalies	17 (28%)	5 (17%)	1 (18%)	0.345	
Cardiac findings	26 (42%)	21 (72%)	3 (27%)	<0.001	

A *p*-Value < 0.05 was assumed to be statistically significant. The *p*-values refer to the association between the delta MCA-PSV and the MCA-PSV >1.5/<1.0 MoM Group. MoM, multiples of the median; MCA-PSV, middle cerebral artery peak systolic velocity; f/m, female/male; Hb, Hemoglobin.

## Data Availability

All data generated or analyzed as part of this study are included in the long version of this paper. This article provides a compilation of the main results and findings. The long version of this work has not yet been published. If required, the complete work can be consulted at any time. If you have any questions, please contact the main authors de Sainte Fare or Axt-Fliedner.

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
