# Peer review of "The Value of Delta Middle Cerebral Artery Peak Systolic Velocity for the Prediction of Twin Anemia-Polycythemia Sequence—Analysis of a Heterogenous Cohort of Monochorionic Twins"

_jcm, 2022, doi:10.3390/jcm11247541_

Round 1
Reviewer 1 Report
The manuscript: ‘The value of delta middle cerebral artery peak systolic velocity for the prediction of Twin Anemia-Polycythemia Sequence — Analysis of a heterogenous cohort of monochorionic twins’ presents results of prenatal difference in the flow velocity of the middle cerebral artery (delta MCA-PSV) assessment and hemoglobin difference measured postnatally in a large cohort of 348 monochorionic diamniotic (MCDA) twins.
The main aim of the research was to analyze the diagnostic potential of Twin Anemia-Polycythemia Sequence (TAPS) by delta MCA-PSV >0.5 MoM in MCDA twin pairs. The topic was analysed previously by various researchers and variable cut-off values for MCA-PSV and delta MCA-PSV were proposed. Presented research is a confirmation of earlier findings on a large cohort of MCDA twins, what is valuable impact in the topic.
The conclusions are generally consistent with the main question of the research and adequate. However, despite original character of the study and a large cohort of tested MCDA pregnancies, there are some week points of the manuscript, which require major revisions.
1. In the method section the diagnostic postnatal criterion for TAPS confirmation was established as inter-twin hemoglobin difference >80 g/l. However, in delta MCA-PSV > 0.5 MoM group, the avarage hemoglobin difference was 75 g/l (for example – Table 3). The reason of this discrepancy should be checked and the manuscript should be corrected.
2. ‘The difference in birth weight of the twins in this group was 433 g (22 %) on average, higher than that of 186 the prenatally diagnosed TAPS cases’ (line 185-187). The sentense is false as the difference in birth weight was 22%, so lower than in MCA-PSV >1.5/<1.0 MoM group. The sentence should be corrected.
3. The manuscript: ‘Performance of Antenatal Diagnostic Criteria of Twin-Anemia-Polycythemia Sequence’ by Liu, et al. published in 2020 in ‘Journal of Clinical Medicine’ regards application of two different cut-off values of delta MCA-PSV in prediction of TAPS and its postnatal course. The manuscript should be included to the references.

Author Response
Dear Sir or Madam
Thank you for taking the time to read our work. I would also like to thank you for your valuable comments.
Reviewer1:
Point 1: In the method section the diagnostic postnatal criterion for TAPS confirmation was established as inter-twin hemoglobin difference >80 g/l. However, in delta MCA-PSV > 0.5 MoM group, the avarage hemoglobin difference was 75 g/l (for example – Table 3). The reason of this discrepancy should be checked and the manuscript should be corrected.
Response 1: In the 61 cases indicated to have TAPS by the delta group prenatally, the average Hb difference was 75 g/l. Not all these 61 cases have been confirmed postnatally by the obvious too low Hb difference. Of the 61 cases that prenatally indicated TAPS, only 29 were confirmed postnatally (see Figure 1). However, the Hb-difference refers to all cases that prenatally indicated TAPS and not only those confirmed postnatally. I have added in the document: "All these data refer to cases that were prenatally indicative of TAPS and not to postnatally confirmed TAPS cases.“
Point 2: ‘The difference in birth weight of the twins in this group was 433 g (22 %) on average, higher than that of 186 the prenatally diagnosed TAPS cases’ (line 185- 187). The sentense is false as the difference in birth weight was 22%, so lower than in MCA-PSV >1.5/<1.0 MoM group. The sentence should be corrected.
Response 2: The weight discrepancy in the group of prenatally undetected TAPS cases was 433g, which is higher than in the delta and cut-off groups (361g, 332g). To reduce confusion with the percentages, I have removed them. Thank you for pointing this out.
Point 3: The manuscript: ‘Performance of Antenatal Diagnostic Criteria of Twin- Anemia-Polycythemia Sequence’ by Liu, et al. published in 2020 in ‘Journal of Clinical Medicine’ regards application of two different cut-off values of delta MCA-PSV in prediction of TAPS and its postnatal course. The manuscript should be included to the references.
Response 3: Thank you for pointing this out. I have included the specified source in the discussion section line: 337-358.
Revised manuscript attached
Yours sincerely
Anthea de Sainte Fare

Reviewer 2 Report
It is very interesting research about TAPS in MCDA twin pregnancies, suggesting delta middle cerebral artery-peak systolic velocity (MCA-PSV) > 0.5 multiples MoM) (delta group) detects more TAPS cases than the traditional method MCA-PSV cut off levels of > 1.5 and < 1.0 MoM. The main study group was 90 cases from retrospective analysis of 348 live-born MCDA twin pregnancies from 2010-2021 year from 5 german centers. I do agree with conclusions, but I would be very happy to see also cardiac findings in all 3 groups: control, "delta" and "old" groups.
The prevalence of cardiac findings is mentioned in table 3 and it was 42% and 72% - so quite high, it would be very interesting to have a separate table, just with these findings. Also, it would be interesting to see cardiac details in 11 cases that were missed by prenatal evaluation (Fig.1)
Author Response
Dear Sir or Madam
Thank you for taking the time to read our work.
Reviewer 2:
Point 1: I do agree with conclusions, but I would be very happy to see also cardiac findings in all 3 groups: control, "delta" and "old" groups. The prevalence of cardiac findings is mentioned in table 3 and it was 42% and 72% - so quite high, it would be very interesting to have a separate table, just with these findings. Also, it would be interesting to see cardiac details in 11 cases that were missed by prenatal evaluation (Fig.1)
Response 1: You are absolutely right that it would be a valuable and interesting additional information. However, we do not have enough detailed information on cardinal diseases and their further process. Therefore, we decided not to investigate them further in this study. We wanted, especially in this paper, to focus on the prenatal diagnostic criteria of TAPS and not to create as many side analyses. We are very happy to try to analyze this aspect further in the PhD thesis. Thank you very much for the inspiring comment, we hope you can understand our concerns.
Kind regards
Anthea de Sainte Fare
Round 2
Reviewer 1 Report
The manuscript may be accepted in present form.